# Plasma CAF22 Levels as a Useful Predictor of Muscle Health in Patients with Chronic Obstructive Pulmonary Disease

**DOI:** 10.3390/biology9070166

**Published:** 2020-07-15

**Authors:** Rizwan Qaisar, Asima Karim, Tahir Muhammad

**Affiliations:** 1Basic Medical Sciences, College of Medicine, University of Sharjah, Sharjah 27272, UAE; akarim@sharjah.ac.ae; 2Department of Physiology & Cell Biology, University of Health Sciences, Lahore 53720, Pakistan; 3Department of Biochemistry, Gomal Medical College, Dera Ismail Khan 29050, Pakistan; drtahir82@gmail.com

**Keywords:** biomarkers, COPD, muscular atrophy, skeletal muscle, spirometry

## Abstract

Skeletal muscle dysfunction and reduced physical capacity are characteristic features of chronic obstructive pulmonary disease (COPD). However, the search for a reliable biomarker to assess muscle health in CODP remains elusive. We analyzed the course of hand-grip strength (HGS) and appendicular skeletal mass index (ASMI) in COPD in relation to spirometry decline and plasma extracellular heat shock protein-72 (eHSP72) and c-terminal fragment of agrin-22 (CAF22) levels. We evaluated male, 62–73 years old patients of COPD (*N* = 265) and healthy controls (*N* = 252) at baseline and after 12 and 24 months for plasma biomarkers, spirometry and HGS measurements. HGS declined significantly over time and plasma CAF22, but not eHSP72 levels, had a significant negative association with HGS and ASMI in COPD. Plasma CAF22 also had an association with walking speed and daily steps count in advanced COPD. Lower ASMI was associated with reduced HGS at all time-point. Narrow age-span of the study cohort and exclusion of lower-limb muscles from the analysis are limitations of this study. Taken together, we report that the plasma CAF22 may be a useful tool to assess muscle weakness and atrophy in COPD patients.

## 1. Introduction

Chronic obstructive pulmonary disease (COPD) is a chronic debilitating disease characterized by dysfunction of multiple systems in addition to compromised lung function [1]. The progressive decline of skeletal muscle is a common manifestation in COPD and is an important predictor of reduced quality of life in COPD. Muscle weakness exacerbates the already compromised exercise capacity in COPD patients, which further diminishes the quality of life. The progressive loss of muscle mass and strength in COPD has been shown to predict morbidity and mortality irrespective of lung function [2]. Skeletal muscle detriment also affects the exercise capacity and degree of functional debility (e.g., walking speed), which show a significant association with disease severity in COPD [3]. Hand-grip strength (HGS) is recognized as a basic measure of determining upper extremities musculoskeletal strength [4] and several studies have reported a decline of HGS in COPD patients [5,6]. We have recently shown a positive association between HGS and forced expiratory volume (FEV_1_%) in COPD [7], which can affect disease outcomes such as hospitalization, exacerbation and risk of death. However, accurate prediction of disease outcomes requires progressive tracking of lung and skeletal muscle dysfunction at various stages of COPD.

Muscle wasting is a common occurrence in COPD. However, accurate assessment of muscle wasting is technically challenging and requires imaging techniques that are costly, time-consuming and may require the radiations exposures [8]. The use of bioelectrical impedance analysis (BIA) is emerging as a popular alternative to radiography due to its easy use and cost-effectiveness [9].

Measurement of plasma biomarkers can be a reliable tool to assess skeletal muscle health, however, search for a reliable biomarker to accurately detect muscle mass in COPD remains elusive. Agrin is a protein released by motor neurons into the synaptic cleft and is required for the clustering of acetylcholine receptors. Proteolytic cleavage of agrin into its breakdown products called C-terminal agrin fragments (CAF) triggers the destabilization of the neuromuscular junction (NMJ) in multiple catabolic conditions. In human plasma, a smaller fragment of agrin called CAF22 is identified. It has been shown that elevated plasma levels of CAF22 can be a potential marker for sarcopenia [10,11] and muscle wasting in other debilitating conditions [12,13,14]. A negative association between agrin expression and the severity of COPD has been described as the patients with moderate to severe COPD show reduced agrin levels than the mild cases and healthy controls [15]. However, despite the well-recognized protective effects of agrin in lungs, its association with muscle wasting and weakness in COPD is not well defined.

The extracellular heat shock protein-72 (eHSP72) plays a pivotal role in stress response and immune regulation [16]. The elevated plasma eHSP72 levels have been associated with the sarcopenia phenotype in aging [17]. However, its association with the loss of muscle mass and strength in COPD has not been investigated. While measuring serial measurements of plasma CAF22 and eHSP72 levels can help track disease progression in COPD, their expression has not been rigorously characterized in relation to spirometry and muscle wasting in COPD. Additionally, it is not known whether these biomarkers can be independent predictors of markers of physical performance such as daily step count and walking speed.

We aimed to analyze the course of spirometry and muscle decline in COPD as well as biomarker potential of plasma CAF22 and eHSP72 in predicting disease progression in a cohort of heterogeneous COPD patients. We hypothesized that the plasma expressions of eHSP72 and CAF22 relate to spirometry decline, HGS and measures of exercise capacity in COPD. We tested this hypothesis by prospectively collecting data and analyzing biological samples from participants with various stages of COPD over two years.

## 2. Materials and Methods

### 2.1. Study Design and Participants

Patients and healthy participants were recruited after obtaining the ethical approvals at the University of Health Sciences, Lahore (approval #UHS/ERB/22587/2016) and Gomal Medical College, Dera Ismail Khan (approval #GMC/157/2016). Patients were enrolled in June 2017 and followed every twelve months for two years. Anthropometric data, plasma collection and measurements of body composition and handgrip strength were performed at each time point. Participants were divided into healthy controls (*N* = 252) and COPD groups (*N* = 265) with an age range from 62 to 73 years (Figure 1).

Based on the global initiative for obstructive lung disease (GOLD) classification, COPD participants were further subdivided into GOLD 1 and 2 (*N* = 136) and GOLD 3 and 4 (*N* = 129) subgroups. COPD was defined as FEV_1_%/forced vital capacity (FVC) < 0.7 with persistent respiratory symptoms according to the GOLD guidelines [18]. Subjects with stable COPD were included while those with unstable COPD (infection, exacerbation and/or hospitalization in the past month), arthritis, myopathies and neurological diseases were excluded [19]. Further, impaired renal function can reduce glomerular filtration and increase circulating levels of CAF [20]. However, our study cohort had no clinical or laboratory signs of renal failure. Among the patients with COPD, GOLD stages 1 and 2 were defined as FEV_1_ = 50–80% (*N* = 66 in final analysis) and GOLD stages 3 and 4 as FEV_1_ ≤ 50% (*N* = 58 in final analysis). Body mass index (BMI) was calculated as kg/m^2^. Appendicular skeletal muscle mass (ASM) and fat mass were calculated with the bioelectrical impedance analysis scale (RENPHO, Dubai, UAE). ASM was divided by the square of height to get an appendicular skeletal muscle mass index (ASMI), as described elsewhere [21]. Five participants dropped out during the study due to exacerbation or death and were excluded from the final analysis. Written informed consent was obtained from all study participants. This study was conducted in accordance with the declaration of Helsinki [22].

### 2.2. Hand-Grip Strength

Hand-grip strength was measured by a digital handgrip dynamometer (CAMRY, South El Monte, CA, USA) as described before [7]. The participants were instructed to sit down with their elbows flexed at an angle of 90° with the dynamometer in hand in the supine position. The participants were then asked to squeeze the dynamometer with maximal strength in a smooth manner without rapid jerking or wrenching. No other body movement was allowed during the procedure. Three attempts were performed with each hand with a 60-s rest between each attempt and the highest value was recorded for analysis.

### 2.3. Spirometry

The FEV1 and FVC were measured using a portable spirometer (Contec SP10, Shanghai, China), according to standards set by the American Thoracic Society [23]. The participants were instructed to inhale maximally until the lungs were full, followed by forceful exhalation into the spirometer until no air could be exhaled [24]. This was done for a minimum of three times and the severity grading was based on FEV_1_% of predicted values according to the GOLD criteria into GOLD 1–4 [25].

### 2.4. Measurement of Plasma Biomarkers

For the analysis of plasma biomarkers, 64–68 participants from the healthy controls and each of the two subgroups of COPD participants were randomly selected. Plasma was assayed using ELISA kits for eHSP72 (ADI EKS-715; Enzo Life Sciences, Inc. New York, NY, USA) and CAF22 (NTCAF, ELISA, Neurotune, Schlieren-Zurich, Switzerland) according to the manufacturer’s instructions.

### 2.5. Statistical Analysis

Anthropometric measurements of the participants were presented using mean and standard deviation as data met the assumption for normality using a Chi-square normality test. An analysis of variance was used to compare continuous variables and the chi-square test was used to determine categorical variables between the groups. Pearson correlation was employed to determine the strength of the relationship between hand-grip strength (dominant and non-dominant) and the lung function (FEV_1_% and FVC). A *p*-value < 0.05 was statistically significant and value of <0.001 as highly statistically significant.

## 3. Results

### 3.1. Characteristics of the Participants

Basic characteristics of the study population are summarized in Table 1.

There was no significant difference in the BMI, fat mass and appendicular skeletal muscle mass among the three groups. Participants with COPD had significantly reduced HGS, walking speed and step count than healthy controls. After adjustment for appendicular muscle mass, which can influence HGS [26], the adjusted HGS was significantly lower in both subgroups of COPD participants (*p* < 0.05), when compared to healthy controls. The COPD participants also showed a significant reduction in adjusted HGS at the 24-month time, compared to baseline. COPD participants with GOLD stages 3 and 4 also had reduced walking speed and a daily step count than healthy controls at all time points. The plasma levels of eHSP72 and CAF22 were significantly higher in the GOLD 3 and 4 subgroup than healthy controls. Participants with the GOLD 1 and 2 subgroup also showed increased plasma CAF22 but not eHSP72 when compared to healthy controls.

### 3.2. Correlation of Biomarkers with FEV_1_%

We investigated the association of plasma biomarkers with spirometry performance in COPD. We reported a modest association between plasma eHSP72 and FEV_1_% in the three groups at various time points. However, a statistically significant association was only found at 24-months in GOLD 1 and 2 (r^2^ = 0.122, *p* < 0.05) and GOLD 3 and 4 subgroups (r^2^ = 0.081, *p* < 0.05). On the other hand, plasma CAF22 shows a relatively stronger association with FEV_1_% at GOLD 3 and 4 at all-time points. When all the participants were pooled together, CAF22 showed significant association with FEV_1_% at baseline (r^2^ = 0.273, *p* < 0.001), 12 (r^2^ = 0.389, *p* < 0.001) and 24-month time points (r^2^ = 0.283, *p* < 0.001). On the other hand, in the pooled analysis, a significant association of plasma eHSP72 with FEV_1_% was only found at baseline (r^2^ = 0.155, *p* < 0.05) and at 12-month time point (r^2^ = 0.094, *p* < 0.05; Table 2).

### 3.3. Relationship of Plasma Biomarkers with Hand-Grip Strength and ASMI

For plasma analysis, a total of 598 samples from three groups of participants were collected at three different time points twelve months apart (*N* = 64–68/group/time point). Six samples were discarded because the eHSP72 was not detected in them. The participants in each group were matched for the disease severity based on the FEV_1_% score. Plasma eHSP72 levels showed a significant association with HGS at 24-months (r^2^ = 0.038, *p* < 0.05) but not at the baseline and 12-month time points. On the other hand, plasma CAF22 maintained a significant association with HGS at the baseline (r^2^ = 0.373, *p* < 0.001), 12-month (r^2^ = 0.31, *p* < 0.001) and 24-month (r^2^ = 0.379, *p* < 0.05) time points (Figure 2).

In addition to muscle weakness, muscle wasting is also a common occurrence in chronic advanced COPD. We investigated the association plasma biomarkers with ASMI as a measure of muscle mass. The plasma eHSP72 levels showed a significant association with ASMI at baseline only (r^2^ = 0.027, *p* < 0.05). On the other hand, plasma CAF22 levels maintained significant negative associations with ASMI at baseline (r^2^ = 0.156, *p* < 0.001), 12-months (r^2^ = 0.139, *p* < 0.05) and 24-months (r^2^ = 0.283, *p* < 0.05; Figure 2).

### 3.4. Relationship of Plasma Biomarkers with Walking Speed and Daily Step Count

We evaluated walking speed and daily step count as measures of physical capacity. Plasma eHSP72 levels had a mild to modest association with walking speed in healthy controls and COPD participants at various time points. However, when the participants were pooled together, it showed a significant association with walking speed at the baseline and 12-month time points. On the other hand, plasma eHSP72 levels showed no association with the daily step count in healthy controls and COPD participants, except in the pooled analysis at the 24-month time point (Table 3).

Plasma CAF22 levels were modest to significant predictors of walking speed and daily step count in healthy controls and participants with GOLD stages 1 and 2. When all three groups were pooled together, plasma CAF22 showed a significant association with walking speed and daily step count at all three time points (Table 3).

### 3.5. Relationship of Hand-Grip Strength with ASMI

Since muscle strength is partly determined by the muscle mass, we investigated the relation between appendicular skeletal muscle index (ASMI) and HGS at various time points. HGS maintained a significant positive association with ASMI at the baseline (r^2^ = 0.098, *p* < 0.001), 12-month (r^2^ = 0.121, *p* < 0.001) and 24-month (r^2^ = 0.195, *p* < 0.001) time points indicating a parallel reduction in ASMI with muscle weakness (Figure 3). We did not find a significant difference between healthy controls and COPD participants when the relationship between HGS and ASMI was analyzed.

## 4. Discussion

Our analysis aimed to predict the longitudinal course of HGS and physical performance and their possible predictors in a cohort of COPD patients. We found a significant decrease in HGS over time in COPD as patients with advanced disease had a more pronounced reduction in HGS. The plasma levels of CAF22 but not eHSP72 showed a significant negative association with HGS and ASMI in COPD. Plasma CAF22 also maintained a strong association with walking speed and daily step count in the advanced stages of COPD. On the other hand, plasma eHSP72 levels showed a modest association with spirometry performance in COPD.

Several studies have investigated candidate biomarkers of lung and muscle health in COPD as a tool to monitor disease severity in COPD, which can be helpful in diagnosing and predicting the disease progress and response to therapy [7,27,28,29]. Our results demonstrated the clinical utility of plasma CAF22, which we proposed as a feasible, noncomplex marker of the muscle decline at various stages of COPD.

The muscle wasting is a major comorbidity in COPD and is experienced by ≈40% of patients with COPD [30]. The loss of muscle mass and strength in these patients contributes to the reduced physical activity and functional dependency, which further exacerbates COPD. While the etiology of muscle wasting is multifactorial, the degeneration of NMJs is a major contributor in multiple catabolic conditions [31]. Agrin plays an active role in the NMJ maintenance by post-synaptic clustering of acetylcholine receptors [32]. Cleavage of agrin by neurotrypsin leads to the release of CAF22 in the plasma. Consequently, with the disappearance of agrin, NMJ is disrupted, which triggers downstream catabolic pathways resulting in muscle wasting. Our findings show that plasma CAF22 is a sensitive circulating marker of muscle wasting and its pathological consequences of the muscle weakness and reduced functional capacity in COPD. An elevation of plasma CAF22 levels has been reported in catabolic conditions such as sarcopenia [10], stroke [13] and heart failure [12]. Using the mouse models, we have previously shown that the degeneration of NMJ contributes to reduced muscle strength in addition to its well established effects on muscle mass [33,34]. In agreement with these findings, plasma CAF22 levels show a direct association with HGS as stroke patients with weaker HGS have higher plasma levels of CAF22. Interestingly, physical rehabilitation partially restores HGS and reduces the plasma CAF22 levels, which suggest reactive recovery of NMJ in stroke patients [13]. Here, we extended these findings to patients with COPD and show that plasma CAF22 levels maintained a significant association with HGS and appendicular skeletal muscle mass in COPD. Degeneration and the blunted reinnervation of NMJs have been reported in muscle biopsies from COPD patients [35], which is probably the major contributor to elevated plasma CAF22 levels and muscle detriment in COPD. In addition to HGS, lower limb strength has also been described as a marker of functional dependence in the elderly. However, the degree of muscle wasting varies between upper and lower limbs in muscle catabolic conditions. We did not investigate the lower limb strength in this study, however the significant correlation of plasma CAF22 with HGS and walking speed in our cohort shows that CAF22 could be a candidate biomarker of muscle decline in COPD. In agreement with this finding, an association between the HGS and walking speed has been demonstrated in the elderly population with sarcopenia [36]. We have previously shown an association between HGS and FEV_1_% in COPD [7]. Thus, the reduced physical activity in our patients with advanced COPD can be partly due to the poor oxygenation in addition to muscle atrophy and weakness. In support of this, the coupling between increased plasma CAF22 levels and reduced peak VO2 and reduced exercise capacity has previously been described in patients with cardiac failure [12]. A strict characterization of association between plasma CAF22 levels and FEV_1_% has not been performed before. While we could not find a significant correlation between CAF22 and FEV_1_% in this study, the modest association between plasma CAF22 and FEV_1_% in our COPD cohort shows that the spirometry decline in COPD can be partly independent of its systemic manifestations.

The plasma eHSP72 is involved in the regulation of immune response and cellular stress. However, chronically heightened eHSP72 in plasma has pathological consequences on multiple tissues including skeletal muscle. Elevated plasma levels of eHSP72 are associated with several diagnostic criteria of sarcopenia including reduced ASM, HGS and walking speed irrespective of age, gender and comorbidities [17]. Interestingly, resistance exercise in the elderly restores muscle mass and reduces the plasma eHSP72 levels further eliciting the negative association between muscle mass and plasma eHSP72 levels [37]. Elderly with higher plasma eHSP72 also show reduced functional capacity [17], which is in agreement with our finding of lower step count and walking speed in patients with increased plasma eHSP72 levels. This association of plasma eHSP72 levels with muscle detriment is independent of plasma inflammatory cytokines levels [17,38]. We found a modest relationship between plasma eHSP72 and FEV_1_%, which became stronger in the advanced stages of COPD. Plasma heat shock proteins levels are shown to increase in pulmonary decline and show a negative correlation with FEV_1_% [39]. The increase in circulating eHSP72 in the advanced stages of COPD is likely a systemic reflection of stress and inflammatory status and contributes to the pulmonary decline in COPD.

Our analyses are subject to some methodological issues that can influence our results. Like other cohort studies, selective survival before the recruitment into the study cohort has to be taken into account. The findings in the study population of 62–73 years may not represent other age groups. We did not measure the lower extremities’ strength, which is an important determinant of quality of life in aging [40].

## 5. Conclusions

In conclusion, we have shown that the expression of plasma CAF22 shows moderate to strong association with muscle mass, force and physical performance in COPD. On the other hand, plasma eHSP72 levels were poor predictors of muscle decline in COPD. Our results revealed that plasma CAF22 could be a potential biomarker of muscle health and physical independence in patients with COPD. Further studies are required to evaluate the efficacy of these biomarkers in monitoring disease progression in COPD.

## Figures and Tables

**Figure 1 biology-09-00166-f001:**
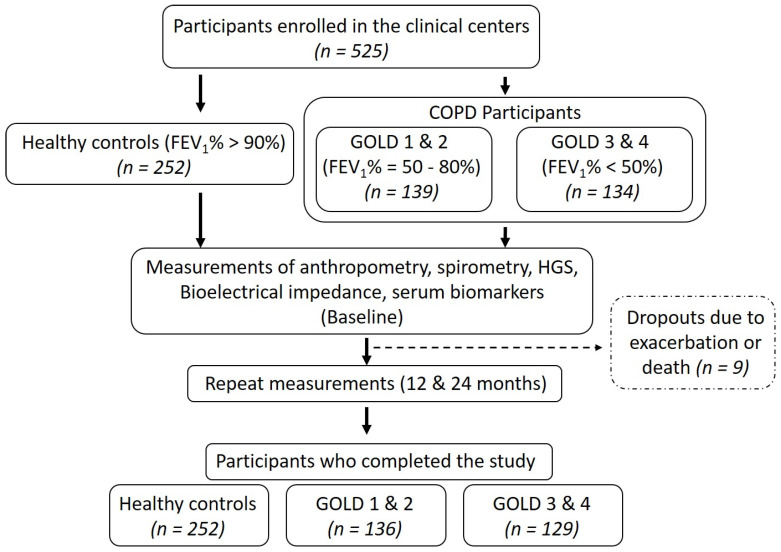
Study flow. FEV_1_%: Forced expiratory volume in the first second and GOLD: global initiative for obstructive lung diseases.

**Figure 2 biology-09-00166-f002:**
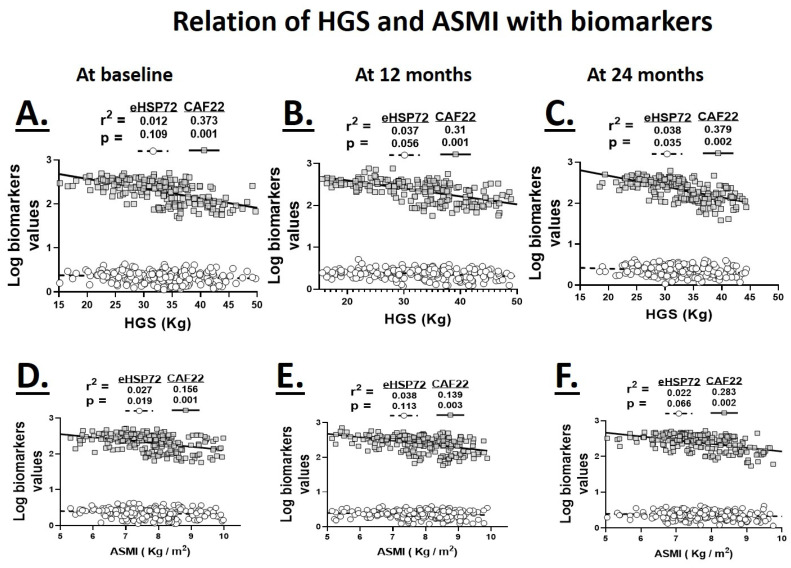
Relationship of serum levels of eHSP72 and CAF22 with hand-grip strength (HGS; **A**–**C**) and appendicular skeletal muscle mass (ASMI; **D**–**F**) at the baseline (**A**,**D**), 12 (**B**,**E**) and 24-month (**C**,**F**) time points in healthy controls and patients with COPD GOLD stages 1 and 2 and 3 and 4 (*N* = 64–68 participants/group/time point for pooled data).

**Figure 3 biology-09-00166-f003:**
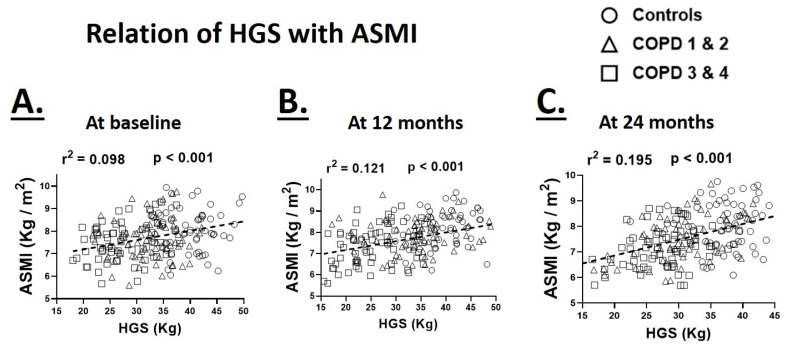
Relationship of hand-grip strength (HGS) with appendicular skeletal muscle mass (ASMI) at the baseline (**A**), 12 (**B**) and 24-month (**C**) time points in healthy controls and patients with COPD GOLD stages 1 and 2 and 3 and 4 (*N* = 64–68 participants/group/time point for pooled data).

**Table 1 biology-09-00166-t001:** Body composition, physical parameters and plasma biomarkers in healthy controls and chronic obstructive pulmonary disease (COPD) participants.

	Non-COPD	COPD GOLD 1 and 2	COPD GOLD 3 and 4
Age at Baseline (Years)	66.4 ± 4.7	67.3 ± 5.2	69.1 ± 4.8
**Body Composition**			
BMI (Kg/m^2^)	(0) 26.2 ± 5.3	(0) 25.4 ± 4.5	(0) 24.6 ± 4.3
	(12) 26.1 ± 5.1	(12) 25.1 ± 4.9	(12) 25 ± 5.2
	(24) 26 ± 4.3	(24) 24.4 ± 4.7	(24) 24.8 ± 4.4
ASM (Kg)	(0) 23.5 ± 4.3	(0) 22.6 ± 3.7	(0) 22.4 ± 4
	(12) 22.8 ± 5.1	(12) 22 ± 4.8	(12) 22.5 ± 4.1
	(24) 22.1 ± 4.3	(24) 22.1 ± 5.1	(24) 21 ± 3.7
ASMI (Kg/m^2^)	(0) 8 ± 1.5	(0) 7.9 ± 1.3	(0) 7.8 ± 1.2
	(12) 7.9 ± 1.6	(12) 7.7 ± 1.4	(12) 7.8 ± 1.1
	(24) 8.1 ± 1.4	(24) 7.6 ± 0.9 *	(24) 7.5 ± 0.8 *
Percent Fat	(0) 41 ± 6.2	(0) 40.6 ± 5.4	(0) 41.3 ± 6.4
	(12) 42.8 ± 6.4	(12) 42.4 ± 6.1	(12) 43.6 ± 5.6
	(24) 43.8 ± 5.8	(24) 44.1 ± 4.3	(24) 45.2 ± 6.4
**Physical Parameters**			
HGS (kg)	(0) 41.5 ± 5.4	(0) 36.4 ± 4.3 *	(0) 29.6 ± 5.2 * #
	(12) 38.4.8 ± 6.4	(12) 33.1 ± 6.3 *	(12) 25.5 ± 4.5 * #
	(24) 37.4 ± 4.3 α	(24) 29.4 ± 5.1 * α	(24) 22.1 ± 4.1 * # α
HGS/ASM	(0) 1.76 ± 0.3	(0) 1.61 ± 0.14 *	(0) 1.34 ± 0.23 * #
	(12) 1.68 ± 0.16	(12) 1.5 ± 0.15 *	(12) 1.13 ± 0.18 * # α
	(24) 1.55 ± 0.21 α β	(24) 1.33 ± 0.13 * α	(24) 1.05 ± 0.11 * # α
10 min Walking Speed (m/s)	(0) 1.18 ± 0.28	(0) 1.09 ± 0.2	(0) 1.01 ± 0.28 *
	(12) 1.15 ± 0.26	(12) 1.02 ± 0.19 *	(12) 0.95 ± 0.26 *
	(24) 1.1 ± 0.22	(24) 1.01 ± 0.17	(24) 0.91 ± 0.22 * # α
Daily Steps Count	(0) 7787 ± 1037	(0) 6373 ± 1098	(0) 4252 ± 636 * #
	(12) 7598 ± 985	(12) 5943 ± 949 *	(12) 3673 ± 512 * #
	(24) 7614 ± 1014	(24) 5611 ± 869 * α	(24) 3361 ± 496 * # α
**Spirometry**			
FEV_1_%	(0) 96.41 ± 5.7	(0) 64.73 ± 6.3 *	(0) 43.7 ± 5.2 * #
	(12) 95.12 ± 6.4	(12) 59 ± 5.7 *	(12) 41.8 ± 4.1 * #
	(24) 96.54 ± 4.3	(24) 56.34 ± 4.4 * α	(24) 39.3 ± 4.2 * #
PEFR%	(0) 90.91 ± 5.5	(0) 74.83 ± 6.3 *	(0) 53.48 ± 6.4 * #
	(12) 88.82 ± 6.4	(12) 71.31 ± 7.3 *	(12) 51.33 ± 4.1 * #
	(24) 89.37 ± 5.4	(24) 67.43 ± 5.5 * α	(24) 48.43 ± 3.6 * #
**Plasma Biomarkers**			
eHSP72 (ng/mL)	(0) 2.11 ± 0.7	(0) 2.19 ± 0.8	(0) 2.31 ± 0.5 *
	(12) 2.22 ± 0.8	(12) 2.34 ± 0.6	(12) 2.41 ± 0.5 *
	(24) 2.29 ± 0.6 α	(24) 2.38 ± 0.7 α	(24) 2.55 ± 0.6 *
CAF22 (pM)	(0) 88.2 ± 15.4	(0) 229.3 ± 44.3 *	(0) 334 ± 69.1 * #
	(12) 104.3 ± 33	(12) 256.3 ± 65 *	(12) 368.3 ± 83 * #
	(24) 122.8 ± 31	(24) 273.4 ± 44 * α	(24) 389.5 ± 42 * α # β

Values are expressed as mean ± SD; one-way analysis of variance.* *p* < 0.05 vs. the non-COPD group at the same time point; # *p* < 0.05 vs. the GOLD 1 and 2 subgroup at the same time point; α *p* < 0.05 vs. the baseline in the same group at the same time point, β *p* < 0.05 vs. 12-months in the same group. The numbers in parenthesis indicate the measurement time points. (*N* = 129–252/group for body composition, physical parameters and spirometry and 64–68/group for plasma biomarkers).

**Table 2 biology-09-00166-t002:** Correlations coefficients of plasma biomarkers with FEV_1_% at various time points (*N* = 64–68/group/time point). The numbers in parenthesis indicate *p*-values.

		Non-COPD	COPD GOLD 1 and 2	COPD GOLD 3 and 4	All Participants
Baseline	eHSP72	0.015 (0.144)	0.081 (0.061)	0.093 (0.034)	0.155 (0.023)
	CAF22	0.053 (0.064)	0.094 (0.054)	0.126 (0.006)	0.273 (<0.001)
12-Month	eHSP72	0.024 (0.117)	0.145 (0.071)	0.139 (0.053)	0.094 (0.005)
	CAF22	0.035 (0.061)	0.106 (0.051)	0.149 (0.036)	0.389 (<0.001)
24-Month	eHSP72	0.031 (0.196)	0.122 (0.041)	0.081 (0.043)	0.076 (0.066)
	CAF22	0.089 (0.063)	0.137 (0.049)	0.143 (0.035)	0.283 (0.005)

**Table 3 biology-09-00166-t003:** Correlations coefficients of plasma biomarkers with walking speed and daily steps count at various time points (*N* = 64–68/group/time point). The numbers in parenthesis indicate *p*-values.

			Non-COPD	COPD GOLD 1 and 2	COPD GOLD 3 and 4	All Participants
Baseline	eHSP72 CAF22	Walking speed	0.034 (0.074)	0.027 (0.128)	0.026 (0.094)	0.052 (0.042)
		Step count	0.013 (0.127)	0.024 (0.134)	0.041 (0.106)	0.062 (0.055)
		Walking speed	0.223 (0.034)	0.151 (0.053)	0.252 (0.024)	0.305 (<0.001)
		Step count	0.149 (0.067)	0.212 (0.004)	0.241 (0.003)	0.364 (<0.001)
12-month	eHSP72 CAF22	Walking speed	0.038 (0.077)	0.047 (0.113)	0.011 (0.123)	0.082 (0.048)
		Step count	0.035 (0.171)	0.106 (0.091)	0.063 (0.137)	0.089 (0.077)
		Walking speed	0.161 (0.004)	0.246 (0.014)	0.202 (0.013)	0.395 (<0.001)
		Step count	0.217 (0.007)	0.143 (0.003)	0.159 (0.002)	0.263 (<0.001)
24-month	eHSP72 CAF22	Walking speed	0.006 (0.186)	0.019 (0.106)	0.055 (0.087)	0.076 (0.076)
		Step count	0.024 (0.133)	0.037 (0.119)	0.083 (0.085)	0.183 (0.033)
		Walking speed	0.147 (0.003)	0.164 (0.021)	0.123 (0.005)	0.329 (<0.001)
		Step count	0.084 (0.041)	0.135 (0.017)	0.117 (0.022)	0.236 (<0.001)

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
