# Peer review of "Plasma CAF22 Levels as a Useful Predictor of Muscle Health in Patients with Chronic Obstructive Pulmonary Disease"

_biology, 2020, doi:10.3390/biology9070166_

Round 1
Reviewer 1 Report
The manuscript by Qaisar and colleagues describes the association of plasma levels of C-terminal fragment of agrin as well as extracellular heat shock protein-72 and muscle fitness in COPD. The study has been designed and described well and the data are sound and based on well-performed statistical analysis.
Major comments: the manuscript would benefit strongly from addition of a a number of relevant references. Line 55, refer to Groffen et al in Eur. J. Biochemistry (1998) on the protein information. It is relevant because of the specific use of the biochemically interesting C-terminal fragment. There are some studies that should be mentioned as well as they will strenghten the paper: Marzetti, Exp. Gerentology 2014, on hip fractures and CAF, Landi et al, Exp Gerontology, on the CAF relation to sarcopenia. Xiao described in 2018 agrin overexpression in COPD (American Journal of Physiology) and Daryadel published in PLOS One a paper on CAF as a marker for renal dysfunction. Have a look at them and add where appropriate. Last but not least, Stephan and coworkers reported on neurotrypsin activity which is local at the NMJ (FASEB J, 2018). Please add this one, since a remark is made on this without a reference (line 221). As far as eHSP72 is concerned, a reference to Asea (2003, Exerc. Imm Rev) is relevant to add. Note that reference 31 in line 257 is incorrect. The study refers to HSP70, not eHSP72.
Methods: line 119, the use of haptoglobin is not clear in the study nor is it found in the rest of the manuscript. Explain why it is mentioned here and where it is used in the study.
Line 126, a p-value < 0.05 was considered as significant. This is correctly done in almost all analyses. Line 151, also at baseline in the COPD gold 3&4 there is a p-value lower than 0.05. The same holds true for figure 2, in which at 24 months there is a significant association for eHSP72 and hand grip strength, which is not mentioned in the text (please correct lines 163/164; note that also CAF22 has a p value higher than 0.001, which is not correct in the text).
Table 3: please, align measured parameters and data in the rows, for instance, correctly done for baseline eHSP72, but not correctly aligned for CAF22 at baseline. In the table, to my opinion there is an association at 24 months between step count and eHSP72 (p<0.05).
Figure 3: maybe it is worthwhile to mention that there is no difference between controls and COPD patients when looking at the relationship between HGS and ASMI.
Discussion: based on the data and literature, the situation on CAF is very clear and warrants a good conclusion. This is not the case for eHSP72: in the discussion the authors mention that "... is in agreement with our finding of lower step count and walking speed in patients with increased eHSP72 levels" (lines 258/259). However, in the results this is not mentioned as such (lines 183/184), but as said under the remark on Table 3, there is a an association. This point also relates to the title of the manuscript: why not put the emphasis solely on CAF22 since eHSP72 is a bit unclear? It does not have to be changed, but maybe it is good to reconsider it in the light of the conclusion ('eHSP72 are poor predictors') and the literature.
minor remarks: line 221, remove 'the' before neurotrypsin; line 255 change resistant exercise to resistance exercise.
Question to the authors: why is in Table 1 the FEV1% under 'spirometry' mentioned as 'predicted'?
Last remark: the paragraph at lines 265 and further weaken the paper a bit. Since the study looks fine and several of the mentioned issues have been raised accross the text, maybe put less emphasis on this part.
Author Response
Dear Editor,
Attached please find the revised version of our manuscript with the revised title ‘‘Plasma CAF22 levels as a useful predictor of muscle health in patients with chronic obstructive pulmonary disease''. We are happy for the constructive criticism and encouraging comments by the reviewers. The manuscript has been revised in accordance with comments and we believe this has significantly improved the quality of our manuscript significantly. The revised and additional text is in red for quick referencing. Please find our responses to specific comments and details about the revisions below.
Reviewer: 1
The manuscript by Qaisar and colleagues describes the association of plasma levels of C-terminal fragment of agrin as well as extracellular heat shock protein-72 and muscle fitness in COPD. The study has been designed and described well and the data are sound and based on well-performed statistical analysis.
Point 1: Major comments: the manuscript would benefit strongly from addition of a number of relevant references. Line 55, refer to Groffen et al in Eur. J. Biochemistry (1998) on the protein information. It is relevant because of the specific use of the biochemically interesting C-terminal fragment. There are some studies that should be mentioned as well as they will strengthen the paper: Marzetti, Exp. Gerontology 2014, on hip fractures and CAF, Landi et al, Exp Gerontology, on the CAF relation to sarcopenia. Xiao described in 2018 agrin overexpression in COPD (American Journal of Physiology) and Daryadel published in PLOS One a paper on CAF as a marker for renal dysfunction. Have a look at them and add where appropriate. Last but not least, Stephan and coworkers reported on neurotrypsin activity which is local at the NMJ (FASEB J, 2018). Please add this one, since a remark is made on this without a reference (line 221). As far as eHSP72 is concerned, a reference to Asea (2003, Exerc. Imm Rev) is relevant to add. Note that reference 31 in line 257 is incorrect. The study refers to HSP70, not eHSP72.
Response: Thank you for highlighting the relevant references. We have added them in the revised manuscript as Landi et al. (Ref. 11 on P-2, L-60), Marzetti et al (Ref. 14 on P-2, L-60), Xiao et al (Ref. 15 on P-2, L-63 with additional text), Daryadel et al (Ref. 20 on P-3, L-96 with additional text), Stephan et al (Ref. 32 on P-9, L-228) and Asea (Ref. 16 on P-2, L-66, albeit a 2005 study). We have also replaced the Ref-31 with Ref-17 in the revised manuscript on P-10, L-266.
Point 2: Methods: line 119, the use of haptoglobin is not clear in the study nor is it found in the rest of the manuscript. Explain why it is mentioned here and where it is used in the study.
Response: We apologize for the erroneous mention. We also investigated serum haptoglobin levels in these patients; however the data was unremarkable and incomplete, and require further characterization including of haptoglobin isoforms. So we omitted it from this study. We have removed the mentioning of haptoglobin from the revised manuscript on P-4, L-125.
Point 3: Line 126, a p-value < 0.05 was considered as significant. This is correctly done in almost all analyses. Line 151, also at baseline in the COPD gold 3 &4 there is a p-value lower than 0.05. The same holds true for figure 2, in which at 24 months there is a significant association for eHSP72 and hand grip strength, which is not mentioned in the text (please correct lines 163/164; note that also CAF22 has a p value higher than 0.001, which is not correct in the text).
Response: Thank you for pointing this out. We have corrected the information about association between plasma eHSP72 and HGS at 24 month (P-6, L-169-170) and about p value for CAF22 at 24 month (P-6, L-172) in the revised manuscript.
Point 4: Table 3: please, align measured parameters and data in the rows, for instance, correctly done for baseline eHSP72, but not correctly aligned for CAF22 at baseline. In the table, to my opinion there is an association at 24 months between step count and eHSP72 (p<0.05).
Response: We have aligned the row in the table 3. We have also added the information about step count with eHSP72 at 24 month in the revised manuscript (P-7, L-190).
Point 5: Figure 3: maybe it is worthwhile to mention that there is no difference between controls and COPD patients when looking at the relationship between HGS and ASMI.
Response: We have added the information in the revised manuscript on P-8, L-203-204.
Point 6: Discussion: based on the data and literature, the situation on CAF is very clear and warrants a good conclusion. This is not the case for eHSP72: in the discussion the authors mention that "... is in agreement with our finding of lower step count and walking speed in patients with increased eHSP72 levels" (lines 258/259). However, in the results this is not mentioned as such (lines 183/184), but as said under the remark on Table 3, there is a an association. This point also relates to the title of the manuscript: why not put the emphasis solely on CAF22 since eHSP72 is a bit unclear? It does not have to be changed, but maybe it is good to reconsider it in the light of the conclusion ('eHSP72 are poor predictors') and the literature.
Response: Thank you for pointing this out and we have added additional text in the results section (P-7, L-190) to highlight significant association between step count and eHSP72 at 24 month in pooled analysis. We have also revised the title and emphasized on CAF22 alone by omitting eHSP72.
Point 7: minor remarks: line 221, remove 'the' before neurotrypsin; line 255 change resistant exercise to resistance exercise.
Response: We have removed ‘’the’’ (P-9, L-228) and changed resistant with resistance (P-10, L-263) in the revised manuscript.
Point 8: Question to the authors: why is in Table 1 the FEV1% under 'spirometry' mentioned as 'predicted'?
Response: Thanks for pointing this out. It was erroneous and we have removed the word ‘’predicted’’ from table 1.
Point 9: Last remark: the paragraph at lines 265 and further weaken the paper a bit. Since the study looks fine and several of the mentioned issues have been raised across the text, maybe put less emphasis on this part.
Response: The suggestion is well taken and we have shortened the paragraph, removing or rewriting some of the issues discussed there (P-10, L-274-278).
Reviewer 2 Report
This article show the relationship of some physiological variables in patients with COPD disease. It is very interesting the relationship between levels of plasma CAF22 and progression of COPD and loss of muscular functional characteristics. Finally, authors conclude that CAF22 could be a promising biomarker of muscle quality.
Overall comments
In general, the study is well is well designed and technically correct. However, some aspects must be revised, especially statistical analysis.
Moreover some methodologies are not explained. How do you measure appendicular skeletal muscle index (ASMI)? Could be interesting study the relationship of leg muscular mass with plasma biomarkers, instead of ASMI.
Conclusions are based on significant results but regressions values are very low, it is necessary justify better this fact
Specific comments
Explain how do you measure ASMI, in 2. Materials and Methods
Line 123: How do you test/assume normality of variables? Explain which test did you performed.
Line 125 – 127: There is not any explanation about analysis of relationship of AMIS and HGS with biomarkers
Author Response
Attached please find the revised version of our manuscript with revised title ‘‘Plasma CAF22 levels as a useful predictor of muscle health in patients with chronic obstructive pulmonary disease''. We are happy for the constructive criticism and encouraging comments by the reviewers. The manuscript has been revised in accordance with comments and we believe this has significantly improved the quality of our manuscript significantly. The revised and additional text is in red for quick referencing. Please find our responses to specific comments and details about the revisions below.
Reviewer: 2
This article shows the relationship of some physiological variables in patients with COPD disease. It is very interesting the relationship between levels of plasma CAF22 and progression of COPD and loss of muscular functional characteristics. Finally, authors conclude that CAF22 could be a promising biomarker of muscle quality.
Overall comments
Point 1: In general, the study is well is well designed and technically correct. However, some aspects must be revised, especially statistical analysis.
Response: Thank you for pointing this out. We used one-way ANOVA for group-wise comparison and Pearson correlation to determine the strengths of associations between different variables. We understand that despite mentioning the p values of <0.05 or <0.001 at various places in the tables, we did not specify these as statistically significant vs. highly significant in the method section. We have accordingly, added this information in revised manuscript on P-4, L-132-133. We have also added additional text in the results section (P-6, L-169-170 & 172; P-7, L-190) to highlight the statistical differences mentioned in the tables.
Point 2: Moreover some methodologies are not explained. How do you measure appendicular skeletal muscle index (ASMI)? Could be interesting study the relationship of leg muscular mass with plasma biomarkers, instead of ASMI.
Response: ASMI was measured by dividing ASM by the square of height as described elsewhere (Hou Y et al, PLOS ONE, 2019, PMID: 31344055). We have revised the information and added the reference in the revised manuscript (Ref. 21 on P-3, L-101-102). We did not have the radiographic tools to measure the leg muscular mass alone. However, in our opinion and based on literature, measurement of whole body muscle mass as measured by ASMI is superior to measuring leg muscles alone in assessing muscle wasting conditions. In support of this, ASMI is emerging as a reliable tool to assess muscle wasting in debilitating conditions (Hou Y et al, PLOS ONE, 2019, PMID: 31344055) including COPD (Han Y et al. Medicine, 2019, PMID: 31577733) and sarcopenia (Deer R et al, Clin Nutr, 2020, PMID: 7153986).
Point 3: Conclusions are based on significant results but regressions values are very low, it is necessary justify better this fact
Response: The point is well taken. We have softened the conclusion section (P-10, L-279-284). Further, since the association of eHSP72 with muscle health in COPD is poor, we have omitted it from the revised title.
Specific comments
Point 4: Explain how do you measure ASMI, in 2. Materials and Methods
Response: ASMI was measured by dividing ASM by the square of height as described elsewhere (Hou Y et al, PLOS ONE, 2019, PMID: 31344055). We have revised the information and added the reference in the revised manuscript (Ref. 20 on P-3, L-101-102).
Point 5: Line 123: How do you test/assume normality of variables? Explain which test did you performed.
Response: We used Chi-square normality test for the normality of variables and have added this information in the revised manuscript (P-4, L-128).
Point 6: Line 125 – 127: There is not any explanation about analysis of relationship of AMIS and HGS with biomarkers
Response: The relationship of ASMI and HGS with the biomarkers is explained on P-6, L-166-178. We understand that some of the information from the table was missing in the text so we have added additional text in this sub-section on P-6, L-169-170 & L-172.
Round 2
Reviewer 1 Report
The revised version of the manuscript by Qaisar et al. has improved substantially and recommendations done in the first round were adopted. I have no further comments.
Author Response
Point 1: The revised version of the manuscript by Qaisar et al. has improved substantially and recommendations done in the first round were adopted. I have no further comments.
Response: Thank you for all the input and valuable suggestions which have significantly improved the quality of our manuscript.
Reviewer 2 Report
The authors did follow my suggestion.
The abstract needs to be updated to include some of the limitations and relax the overly strong conclusion that may stem from reliance of p values, which you adequately accounted for with the updated document.
Author Response
Point 1: The authors did follow my suggestion.
The abstract needs to be updated to include some of the limitations and relax the overly strong conclusion that may stem from reliance of p values, which you adequately accounted for with the updated document.
Response: Thank you for the constructive criticism and valuable suggestions which have considerably improved the quality of our manuscript. We have added the study limitations and have softened the conclusion in the abstract.